# Intelligent Assistive Technology Ethics for Aging Adults: Spiritual Impacts as a Necessary Consideration

**Tracy J. Trothen**

School of Religion and School of Rehabilitation Therapy, Queen's University, Kingston, ON K7L 3N6, Canada; trothent@queensu.ca

**Abstract:** Potential spiritual impacts of Artificial Intelligence (AI) driven Assistive Technologies (AT) for older adults are absent in most ethics conversations. Intelligent Assistive Technology (IAT) is the term used to describe the spectrum of Assistive Technologies that use AI. In this theoretical essay, I begin by introducing examples of AT and IAT for older adults with age-related disabilities. I argue that spirituality is a marginalized value in ethics that must be considered if IAT ethics is to address the whole person. Some of the potential spiritual impacts of IATs will be suggested through engagement with three core spiritual needs. I ask how IAT might impact these three core spiritual needs. This is not meant to be an exhaustive study of the spiritual impacts of AT. Through the engagement of one approach to spiritual needs, this article proposes that IAT ethics issues intersect with the spiritual needs of aging adults and, therefore, that potential spiritual impacts ought to be addressed as part of IAT ethics for older adults.

**Keywords:** assistive technology (AT); artificial intelligence (AI); intelligent assistive technology (IAT); spirituality; aging; ethics



## 1. Introduction

Potential spiritual impacts of Artificial Intelligence (AI) driven Assistive Technologies (AT) for older adults are absent in most ethics conversations. Intelligent Assistive Technology (IAT) is the term used to describe the spectrum of Assistive Technologies that use AI. In this theoretical essay, I begin by introducing examples of AT and IAT for older adults with age-related disabilities. I argue that spirituality is a marginalized value in ethics that must be considered if IAT ethics is to address the whole person. Some of the potential spiritual impacts of IATs will be suggested through engagement with three core spiritual needs. I ask how IAT might impact these three core spiritual needs. This is not meant to be an exhaustive study of the spiritual impacts of AT. Through the engagement of one approach to spiritual needs, this article proposes that IAT ethics issues intersect with the spiritual needs of aging adults and, therefore, that potential spiritual impacts ought to be addressed as part of IAT ethics for older adults.

### Assistive Technologies for Aging Adults

Assistive technologies (ATs) for older adults, and particularly for those suffering from physical disabilities or types of dementia, have been proliferating. These technologies help older adults do things that they otherwise could not or that may be difficult to perform. ATs range from emergency help-buttons to cutting-edge bathroom cameras that monitor daily self-care such as tooth-brushing. These cameras are connected to software AI that will speak and remind the person to brush their teeth, or take medications, if they seem to have forgotten (Cook et al. 2020). Other examples of AI that offer assistive functions include popular devices such as Google Home and Alexa. In addition to providing information in response to voice questions, Google and Alexa can turn lights on or off and lock and unlock doors. Voice-controlled alarm clocks can help one to set wake-up times if the clock

buttons are hard to see or the manual process is too complex. Medical alert devices can be worn on one's person and may include a monitor that will alert emergency help if a fall is detected. Button hooks can help do up buttons. Robotic vacuums can help keep living spaces clean. Wheelchairs and walkers can give or improve mobility, allowing someone to walk to the store who otherwise could not. Exoskeletons can restore or improve mobility to working or paralyzed limbs. Bathroom bars and handles can help one to access the shower or tub more safely. Hearing aids amplify sound. Eyeglasses help vision. Locating devices can be used to check on someone's whereabouts, in case they get lost or confused.

And there is more. AI potentially can assist not only with physical needs but also with emotional and spiritual needs. Social companion chat-bots, such as Replika, can provide company, alleviating loneliness (Ta et al. 2020). Paro, and other AI artificial pets, have comforted, reduced anxiety, and increased sociability of isolated aging adults, particularly those adults with dementia (Banks et al. 2008; Johnston 2015). Mindy (a diminutive humanoid-type social robot) is showing promising results in pilot studies as a robot that helps with loneliness. In a pilot study in Canada, residents in retirement communities who experienced Mindy overwhelmingly asked to visit with Mindy again (St. James 2021). Mindy and other humanoid-like robots such as nurse-bot Grace (Cairns 2021) are aiding in healthcare. The COVID pandemic has shown us the harms of isolation, particularly for older adults in long-term care homes or other healthcare facilities.

Other AI-supported AT devices include tablets or iPads that are used to deliver mindfulness apps to help with stress and improve the quality of life for those with mild cognitive impairment (Tran et al. 2020). Digital technologies such as Zoom-type platforms, and immersive tech including augmented and virtual reality games are helping people with dementia to improve their cognitive abilities (Dulau et al. 2019) and connect with communities and individuals. Holographic augmented reality tech is being developed to help people to navigate new or forgotten living spaces. Explicit spirit-focused tech is also emerging. Robots offer blessings, devices such as the Smart Rosary bracelet to track and help with prayers, and even snippets of spiritual guidance (Bettiza 2021; Gibbs 2017; Oladokun 2017). Assistive technology may help support aspects of personhood beyond the physical although such tech supports are not usually included in lists of AT.

In this article, I broaden the meaning of AT to include assistive devices that address multidimensions of being human. I also narrow my focus to the AI subset of ATs that may be used by older adults. Ethical concerns of these technologies include not only the issues that I identify in the next section but also potential spiritual impacts. I will argue that IAT—explicitly spiritual or not—may impact a person's spiritual needs. These potential spiritual impacts are complex and include potential benefits and potential harms that should be considered when choosing what IAT to create, offer and select. My main objective is to demonstrate that IAT has possible spiritual impacts that need to be considered as part of ethical analyses that seek to address the whole person. AI is developing rapidly. The potential spiritual impacts of AI tech, especially on vulnerable groups including aging adults, must become widely recognized as an important aspect of IAT ethics if we are concerned with the person as a whole.

## 2. Approaching IAT Only through a Principlist Bioethics Lens Is Problematic

In ethical examinations of AT, much attention and favour have been given to what have become widely recognized as the four bioethics principles: respect for autonomy, non-maleficence, beneficence, and justice (ex., Martin et al. 2010). These four principles together with iterations of the professional ethics principles of fidelity, veracity, and self-care (Corey et al. 2014) are the principles that are often referenced in ethical assessments of technology and other bioethical and professional ethics topics. While the principles of professional ethics vary across disciplines to a degree, the basic duties of trust, truth-telling, and promise-keeping are common.

While these principles are important and, as we shall see, help to unveil significant ethical issues, a strong focus on these principles obscures important but marginalized

values, often accepting normative understandings of societal values. Traditional bioethical principlist approaches can mask the underlying values that inform accepted interpretations of principles. Socially marginalized values often are not considered in such approaches and the potential impacts of IAT on these values are not explored. Applying only a principlist approach to vulnerable and marginalized older adults who use tech for assistance in living, neglects important aspects of being human, including the spiritual aspect. This neglect of the spiritual dimension may result in the use of IAT that either fails to address core spiritual needs or presents potential harm to these needs.

*A Principlist Approach: Some Key Ethical Issues Regarding AT and Aging Adults*

A principlist approach has directed most attention towards issues most associated with the potential violations of the four traditional bioethical principles. Not only has new tech increased the possibilities for such violations, but new tech is surfacing more challenges to the limitations of a traditional bioethical approach. First, I will identify a few of the important ethical issues raised using a principlist bioethical approach. Next, I consider how a principlist approach has been shaped not only by an accepted list of principles but how these principles have become normatively shaped (and limited) by worldviews that have ignored alternative and marginalized values including spirituality.

I do not claim to provide a comprehensive overview of IAT ethics issues. My point is that while the following ethical issues are very important, the marginalized value of spirituality raises additional important ethical issues. AI options are being developed quickly. AI used in an assistive capacity will have increasing impacts on the whole person, as more IAT is developed and becomes available. There is a need to broaden and deepen ethical analyses of IAT before we begin seeing unanticipated harmful impacts, and to allow us to enhance positive impacts.

One of the main issues driving the creation and use of AT is the desire to help seniors maintain some autonomy by facilitating enhanced independence and to "age in place." The ability to age in place is highly valued and seen as enhancing respect for autonomy by allowing aging adults to stay in the homes in which they have been living for as long as possible. IAT that can be translated successfully from lab to bedside or home (Wangmo et al. 2019) can help make this possible by detecting falls, supporting medication compliancy, monitoring movement and sleep, and assisting with the completion of other daily living activities ranging from dressing to brushing teeth.

Another expression of respect for autonomy is informed consent. Consent requires full disclosure of information, voluntariness, comprehension, and competence. Consent has received a lot of attention in AT ethics. Concern for informed consent has rightly led, for example, to concern about veto controls for people with cognitive impairment including those with dementia. Veto controls (ex., the ability to opt-out of surveillance-based tech at any time) are not always doable options for people with dementia. Also, consent from aging adults to use augmentative communication tech can be difficult to obtain without paradoxically using the tech, sometimes leading to the use of substitute decision-makers. And consent for all aging adults using monitoring devices must be regularly renewed since AI (including programming of their devices) regularly changes. Ethicists also ask regulatory questions such as how can we ensure that consent is regularly updated in ways that ensure a clear understanding of users? And are there appropriate regulatory bodies in place to monitor companies who monitor?

Issues related to surveillance and the privacy of collected personal data are understandable foci in ethical analysis of AT, especially given the increasing role of AI. Questions about the weighing of potential harms (non-maleficence) such as surveillance and privacy infringements against potential benefits (beneficence) such as increased independence are regularly discussed (Cook et al. 2020). The weighing of these potential harms and benefits can be complex, involving reflections on whether self-sufficiency and freedom to live as one chooses to outweigh feelings of vulnerability generated by being monitored, a lack of privacy, the sharing of personal data, and possible uncertainty about the safety or reliability

of the AI being used. Privacy is an issue for IAT that collects personal information with software programs and uses digital networks. The infringement of surveillance technologies on one's privacy has to be weighed against the freedom to continue living where one wishes and, perhaps, to minimize reliance on visible and immediate human support. Given the number of hacking cases, secure data management cannot be completely guaranteed despite the conviction that people have the right to confidentiality and consent regarding personal health information (Martin et al. 2010).

There are also potential negative affective impacts that are beginning to be explored, associated with the use of IAT. For example, there are fears about possible unanticipated usage of data collected from such technology. It may be possible that AI in assistive devices someday could be used to read facial expressions to determine if someone is lying in order to judge them for their beliefs or politics. And IAT can bring other challenges at an emotional and practical level. It may be confusing or disturbing to use the AT, even if the person does not have to do much. For example, hearing a disembodied voice telling you to brush your teeth may be confusing (Cook et al. 2020). Cleaning and changing batteries on a hearing aid can be challenging. All of these things can be stressful and worrisome for users of these technologies.

Justice is the last of the four traditional bioethical principles. Justice, as a bioethical principle, has tended to be most associated with distributive justice and procedural justice. As a result, questions about fair access to technologies, affordability, and how services and resources are assessed and delivered have been the focus of many ethics examinations of IAT when the principle of justice is engaged (Wangmo et al. 2019).

Social justice concerns are gaining more note in IAT bioethical justice conversations. Many IATs are used by aging adults with age-related disabilities. Ageism and discrimination against people with disabilities are two immediate layers of systemic oppression that make many people who use IATs more vulnerable. While it is important to acknowledge that some older adults have much more privilege and power than many others (Holstein et al. 2011), older adults often experience economic hardship, social invisibility, and devaluation. Questions about how machine intelligence might reflect and reinforce bias are beginning to be asked. There is discussion regarding the potential stigmatisation of those who rely on AT including companion bots (European Parliament 2020). Questions of who gets to decide which characteristics are disabilities that need fixing and which are features of human diversity that make us collectively richer are being identified (Brashear 2014). Other justice issues include who should own IATs and AI more generally. Potential labour force worries about replacement by tech in healthcare are surfacing. On the flip side, IATs could support staffing shortages in care. Suffice it to say that the justice side of IAT ethical analysis is gaining complexity as more voices widen the conversation.

Professional ethics principles are also attended to in AT ethics, especially fidelity. Fidelity includes issues related to promise-keeping, integrity, trust, and accountability. Trust grows out of a relationship and is recognized as having an impact on the potential effectiveness of healthcare. Factors including hacking incidents, surveillance, feelings of vulnerability, and an erosion of trust in professionals and government, have generated growing ethical issues. These concerns combined with rapid technological development have contributed to AI governance questions. Globally, regulatory bodies that address professional codes of ethics, procedural regulations, and standards regarding the use of AI are emerging rapidly. Canada launched the first national strategy on AI in March 2017 followed by Japan and then several more countries (European Parliament 2020).

Bioethical and professional ethics principles have helped us to identify some important ethical issues but an exclusively principlist approach can limit our capacities to perceive additional and vital ethical issues (Holstein et al. 2011). These bioethical and professional ethics principles are necessary for raising ethical issues associated with IAT but do not give the whole story. The Panel for the Future of Science and Technology, of the European Parliament (2020), concluded that potential psycho-relational impacts of AI have been neglected in AI ethics. These impacts include psychosocial implications for relationships. For

example, AT can increase feelings of isolation through reduced human contact (Skjuve et al. 2021), and/or increase connection through digital communication. Another study found that human touch is an indispensable good and that IAT should always be a supplement to human care and not a replacement for human care (Wangmo et al. 2019). The European Parliament report authors raise questions about the meaning of human emotional attachments to robots. What if a robot is programmed (either by the creator or a hacker) to financially manipulate and exploit people with whom they have developed relationships? Or if bots are used to convince people to act in ways that they otherwise would not? (European Parliament 2020, p. 18). Research has shown that AI-driven robots can influence people to become less altruistic and more narcissistic (Christakis 2019). While psycho-relational impacts have been neglected in AI ethics, spiritual impacts have been virtually invisible in ethical analyses.

### 3. Beginning with Core Spiritual Needs Shifts the IAT Ethics Conversation

The bioethical principles most used are usually restricted to the four and interpretations of these four tend to be limited. The potential consequences identified often are shaped by these prevalent interpretations. It is important to step back from this foundational set of principles and probe how these principles are interpreted and engaged, and what may be missing. I suggest that particularity of core values and understandings of what it means to be human vary and that if we are to attempt to be more inclusive in ethical examinations of IAT, it is important to consider this diversity and expand ethical examinations of IAT.

Normative North American culture privileges an extreme individualism. A social emphasis on individual rights means that in IAT ethical discussions, rights-oriented values such as security, privacy, surveillance, choice, and independent living are often implicitly assumed and absorbed as some of the most desirable goods (Waters 2015). While these are important goods, they may not always be the most desirable or realistic goods from all perspectives. If we question the over-valuing of individualism, we will ask questions about the possible hyper-valuing of independence including whether aging in place is really the highest or even most desirable good for everyone. Aging in place has been assumed to be the most desirable choice for most adults; independence and self-sufficiency are often uncontested values. But independence and self-sufficiency are illusions that are related to social beliefs about what it means to be strong and of value. Almost everything we do hinges on others. When we eat a meal, most of us buy the food from grocery stores, which rely on the work of many employees. Farmers grow our food. When I order a phone over the internet it does not magically appear in two days. Several people are involved in producing that phone and delivering it to me. While some self-sufficiency is gratifying and desirable for most of us, the belief that we live without relying on others is simply untrue.

Feminist and other intersectional ethicists who are concerned with social marginalization propose a relational approach to autonomy (Agich 2010; Holstein et al. 2011) that is built on our collective interdependence. Many faith and spiritual perspectives assume and celebrate the interconnection of life and human interdependence. A respect for relational autonomy can provide a principlist base for promoting IAT that can help people live in ways that emphasize both community and the individual. A focus on community as an ethics starting point means that responsibility for other life reshapes understandings of individual rights and meaningful choice. For example, a reframing of autonomy as relational autonomy may subtly change the IAT that is designed. If successful aging (Holstein et al. 2011) is less about living on my own and minimizing visible reliance on others, and more about living in ways that encourage transcendent moments and healed relationships (on which I expand later in this article, as aspects of spiritual needs), then IAT that helps us to connect with others and to experience awe through sitting or walking in nature or reading a poem will be prioritized more.

I am not advocating an either/or approach to possible IAT development and use. I am suggesting that it is important to become aware of the core values that inform not only

our choices but also our understandings of ethical principles. Spirituality, as a possible core value, offers alternative views of "successful aging" and respect for autonomy. While spirituality has been researched in relation to occupational therapy (i.e., a form of therapy for recuperation from physical, sensory, or cognitive problems that focuses on rehabilitation through everyday activities sometimes using assistive devices or techniques to address barriers to these needs) (McColl 2011), and there are many ethical examinations of AT, there is a lack of analyses that engage spirituality and IAT ethics. A worldview that includes awareness of spiritual needs reshapes not only how we see these principles but our perception of marginalized values and invisible ethical issues.

Let us move from a consideration of respect for autonomy to another bioethical principle: beneficence. The duty to do good ought to inform most everything that we do. The difficulty is in what it means to do good and who decides what is good. Cautions against paternalism are meant to protect people from having what is perceived as good, particularly in the context of consent and medical treatment, imposed on them by healthcare professionals, against their wishes. While not all paternalism may be unwanted or unconstructive, the clear imposition of a professional's recommendation on a vulnerable client is an abuse of power. Regarding IAT, it is important to ask whose good is at stake in each case. The big 5 tech companies—Google, Facebook, Microsoft, Apple, and Amazon—are often but not always the producers of IAT or their tech components. What is seen as good may be partly shaped by far-reaching economic interests such as the big 5. These interest groups do impose beliefs about what is good and needed by people (including vulnerable people) under the guise of marketing. While purchasing IAT that benefits these companies is not necessarily problematic, it is important to know that sometimes what is presented as good may camouflage other interests.

Taking the issue of what is good a step further, in an age of technological enhancement we need to continually ask what does "better" mean. In theory, most of us would like a better life with better living conditions, even if we do not need more than we have. If we begin an ethical analysis of IAT from a perspective valuing core spiritual needs, the meaning of doing good (beneficence) shifts. Interdependence, not independence, is prioritized. Human fragility and vulnerability are not assumed to be undesirable.

Similarly, one must ask about the meaning of avoiding harms. What we see as harmful shifts depending on one's core values. While the infringement on one's capacity to live alone may be perceived as a highly significant harm by some, for others a greater potential harm may be being lonely or unable to walk in nature or hear music.

Briefly, consider justice. Much bioethical analysis has approached justice as mainly about fairness, access, and allocation of resources. Intersectional approaches emphasize systemic power gaps and marginalization of some groups based on racialization, disability, age, sex, gender, and many other factors. Co-design has emerged as a social justice principle that has moral relevance to IAT. The valuing of care-receivers' self-identified needs and desires means that representative voices be heard at the design stage as well as the implementation stage. As we will see, perceptions of IAT and the older adults with age-related disabilities who use IAT may be coloured by ageism and ableism. Spiritual and other relational needs are important but often unacknowledged and unarticulated considerations when assessing the potential impacts of IAT.

Various modes of ethical reasoning including feminist, relational, and virtue approaches identify marginalized values and worldviews as key aspects of ethical analysis. Deliberate consideration of possible spiritual impacts shifts and expand the IAT ethics conversation.

## 4. Spiritual AIM

The identification of potential spiritual impacts of IAT on older adults requires an understanding of spirituality and spiritual needs. There are different possible approaches to the identification of spiritual needs and the spiritual distress that can result from unmet spiritual needs, including Elizabeth McKinley's model (MacKinley 2017), pastoral theolo-

gian Carrie Doehring's approach (Doehring 2015), psychologist Kenneth I. Pargament's model (Pargament et al. 2017), and various spiritual assessment tools. Also, the various organized religions have ways of identifying spiritual needs mediated by faith convictions.

In this article, I draw on the Spiritual Assessment and Intervention Model (Spiritual AIM) (Shields et al. 2015) developed by Michelle Shields. As a clinical spiritual care supervisor/educator with the Canadian Association of Spiritual Care, I have found this spiritual assessment tool helpful in identifying met and unmet spiritual needs and suggesting a dynamic treatment plan. Spiritual AIM has not been empirically validated but it arose out of years of concrete spiritual care patient experiences, is peer-reviewed (Shields et al. 2015), and is based on an inclusive understanding of spirituality that is not bound to religion. In addition, the understanding of spirituality proposed in the Spiritual AIM is congruent with the 2009 consensus definition of spirituality in palliative care: "Spirituality is the aspect of humanity that refers to the way individuals seek and express meaning and purpose and the way they experience their connectedness to the moment, to self, to others, to nature, and to the significant or sacred" (Puchalski et al. 2009). While this consensus definition was developed specifically for a palliative care context, it encapsulates cross-contextual elements of spirituality.

Additionally, the distillation of three core spiritual needs makes the definition of spirituality proposed in the Spiritual AIM a manageable starting point for a theoretical exploration of the potential spiritual impacts of IAT on aging adults. In the Spiritual AIM, "spirituality (is defined) as encompassing the dimension of life that reflects the needs to seek meaning and direction, to find self-worth and to belong to community, and to love and be loved, often facilitated through seeking reconciliation when relationships are broken" (Shields et al. 2015, p. 78). As will become apparent, these three needs include the aspects identified in the consensus definition of spirituality. I use the Spiritual AIM in this article, not as a diagnostic and treatment tool, but to theorize some of the potential spiritual impacts IAT poses.

To reiterate, the three core spiritual needs outlined in Spiritual AIM that I will consider in relation to potential impacts of IAT are: (1) the need to seek meaning and direction, (2) the need to find self-worth and belonging to community, and (3) the need to love and be loved facilitated by seeking reconciliation in broken relationships. Unmet core spiritual needs tend to emerge more clearly in times of crisis but everyone has all three of these core spiritual needs, any of which may not be well met at any given time. Technology has the potential to exacerbate these core spiritual needs and also to help us better meet these needs. I will reflect briefly on each of these core spiritual needs in relation to the potential impacts of IAT on aging adults. My modest objective is to suggest that there are potential spiritual impacts that ought to be addressed in ethical explorations of IAT.

## 5. Potential Spiritual Impacts

### 5.1. The Need to Seek Meaning and Direction

Humans have a need for meaning and direction. When this need is not adequately met, we can become overwhelmed by a barrage of complex questions including what is valuable about being human and what is valuable about life. Questions of meaning and direction can become more prominent as we age (Friedman 1985). In particular, questions about purpose, identity, and legacy can be expressions of this core spiritual need (Kestenbaum et al. 2021). As we age, as many scholars have pointed out, questions regarding meaning and direction often change (Tornstam 2005). For example, according to Lars Tornstam's theory of positive gaining, older adults may gain a new perspective about loneliness, and find comfort in alone times and a greater sense of meaning in life. So, for some older adults who have matured wisdom, the spiritual need for meaning and direction may be less pronounced. For others who may not have as much experience living and working through crises and other challenges, the spiritual need for meaning and direction may become more pronounced as dying and death become anticipated. In what ways might

IAT help or hinder constructive engagement with these questions, and promote Tornstam's "gerotranscendence" or positive aging?

Part of meaning and direction may be questions about one's relationship to others, and the rest of the world. As philosopher George J. Agich argues, we need to recognize "dependence as a nonaccidental feature of the human condition" (Agich 2010). People who struggle with this core spiritual need may need help to understand that we are connected, that we matter, and that we are not alone. Paradoxically, a desire to be self-sufficient may restrict the support of knowing that life is interdependent. IAT may hinder or help with making sense of these relationships. Monitoring devices may serve as constant reminders of insufficiency or these devices may remind one instead of the benefits of leaning a bit on others. AI "nudges" such as reminders to brush one's teeth or eat a meal may be welcomed as supports that assist people in continuing to live where they wish, or these nudges may be a source of relentless reminders that one is not able in the way one wishes to be. Similarly, digital platforms can serve as assistive devices to connect us with community and loved ones even in the midst of a pandemic, or they may make us feel frustrated or incompetent if we struggle to make them work.

Part of meaning and direction lies in one's relationship to the sacred or transcendent. Philosophers including Charles Taylor, Herbert Dreyfuss, and Sean D. Kelly are convinced that the loss of the transcendent is at the root of "the modern malaise" (Taylor 2007; Dreyfuss and Kelly 2011). If this is so, then experiences of transcendence should be considered an important ethical issue, of a spiritual quality, in IAT for older adults. Philosopher Shannon Vallor similarly writes, from a virtue ethics perspective, that "[t]he unresolved crisis of the 20th century, still with us in the 21st, is a crisis of meaning—the meaning of human excellence, of flourishing, of the good life" (Vallor 2016, p. 247). Psychologist Pargament's research shows that people find the sacred in many things ranging from nature, gardening, music, and sport, to religion. Part of finding the sacred is the experience of what is often understood as spiritual emotions such as awe, elevation, hope, and transcendence (Pargament et al. 2017). If a person finds transcendence in nature, it may be that assistive devices potentially including intelligent exoskeletons or emerging intelligent (smart) wheelchairs could support the core spiritual need of seeking meaning and direction. Or AI-powered hearing supports and potentially bionic eyes.[1]

For other older adults, legacy questions are an important part of finding meaning and direction. Digital devices including extended and augmented reality, and Zoom recordings could help one to pass on messages, photos, and other memories to family, friends, and future generations. At the same time, a growing belief that one cannot navigate this new tech may add to feelings of irrelevance and questions as to whether one has anything of value to pass on to the next generations. How IAT is presented and engaged by care providers and others makes a difference to potential spiritual impacts.

Robots as IATs also pose mixed potential spiritual impacts for older adults seeking meaning and direction. As I discuss in the following section on the need to find self-worth and belonging to community, social chat-bots and interactive robots such as Mindy may mitigate feelings of loneliness. But they may also stir up additional questions of meaning. These questions may not be problematic to ask unless one is already feeling overwhelmed by a sea of questions about meaning and direction. If one leans on a bot for company, does that relationship really count as something of value? Are bots like humans or are they simply collections of well-designed tech that hold no meaning beyond entertainment? Maybe entertainment is valuable in itself but these questions may add to internal angst. Or perhaps a chat-bot may offer responses that support insight or offer comfort. Another issue that is emerging regarding the psycho-social implications of bots is the anthropomorphizing of bots: some people find physical robots eerie since they try to look humanoid but do not quite succeed. This uncanny valley phenomenon may be confusing or frightening (Braxton 2015).

There are sure to be other impacts of IAT on the spiritual need to seek meaning and direction. A big related question is what are we seeking through robotics and other IAT?

Is part of what we are searching for a "something more" that may not be satisfied by technology? It may be that spirituality and spiritual disciplines have more to offer us (or at least something distinctive to offer us) in these quests for meaning than the latest tech (Smith 2022; Mercer and Trothen 2021; Mercer and Trothen 2015). Also, as AI technology changes, so too will the questions it surfaces about the meaning of our lives as we age. Some people will be stimulated or comforted by the supports offered by IAT while others will find the technologies confusing or overwhelming by adding to these already big questions.

### 5.2. The Need to Find Self-Worth and Belonging to Community

The need to find self-worth and belonging to a community was the most frequently reported spiritual need in a 2021 retrospective analysis of 376 patient encounters in a six-month period at an outpatient palliative care unit of a US hospital (Kestenbaum et al. 2021). This core spiritual need is about relationships with self and others. Many IATs have the potential for enhancing the user's self-worth and improving their connections to community. However, there are also potentially harmful impacts from IATs on this spiritual need.

Social bots such as Mindy, which is used in Canadian retirement communities, show promise in alleviating loneliness and enhancing fun for those who engage with the bot (St. James 2021). Talking robots may improve mental health and lessen loneliness for seniors in care homes (Kolirin 2020). Social bots may also increase one's sense of self-worth and may help some users to risk engaging more in their human relationships (Ta et al. 2020; Skjuve et al. 2021). But bots also have potential negative impacts on the core spiritual need of self-worth and belonging that should be considered in ethical examinations of the use of bots as IATs. Among these potential harms are stigmatization, social isolation, deception, a lack of mutual relationships, and the reinforcement of systemic marginalization. Some of these potential benefits and harms can be extended to other IAT besides bots.

Replika is an example of a widely used social chat-bot that is decreasing loneliness and helping users feel better about themselves. Marketed as the "AI companion who cares. Always here to listen and talk. Always on your side," Replika is designed to be a non-judgmental "friend" to users (Replika 2021). Social chat-bots such as Replika are gaining usage. Initial studies indicate that Replika increases human connection for some users but increases social isolation for others (Skjuve et al. 2021), with these latter users relying increasingly on Replika for relationships and less on humans.

Some studies have found that people who use bots, including physical bots and avatars or other digital bots, may be stigmatized for their "strange" companions (Skjuve et al. 2021). Additionally, the uncanny valley phenomenon (Braxton 2015) may exacerbate the potential for stigmatization, which could result in increased social isolation from some human relationships. Visible IAT can increase discriminatory attitudes and stigmatization. In a quest to help connect people more easily with community, there is the perverse risk of tech widening this social gap.

In addition to the stigmatization of relationships with social bots, humanoid-type bots like Mindy, or pet bots, there are other potential harms associated with emotional attachment to a machine. An issue that is being raised by some ethicists is the potential deception involved in users who come to believe that they have a mutual relationship and genuine shared feelings with a bot (Winfield 2020). One response to this critique is that if users are informed that their bot is a machine (not a human or real animal), then any deception is imaginative self-deception, which is morally different from other-deception (Weber-Guskar 2021). Furthermore, deception sometimes can be justified on the grounds of beneficence. While these relationships may not be "real," in that they lack the mutuality of a human-human relationship, some ethicists question whether all "good" relationships must be mutual (Wangmo et al. 2019). It may be that an aging adult gains a sense of company and enjoys talking to a monitoring device that "speaks" to them to remind them to do everyday tasks or to take medications on time. A belief in the realness of relationships with IAT may also assist in positive outcomes such as the promotion of healthy eating and other

self-care practices. This "relationship" may have a good impact on the spiritual need for community, so long as this device does not replace community, resulting in greater social isolation. Deception may not be a harmful consequence of IAT unless the IAT becomes seen as one's central or even only relationship, which could result in a deeper sense of aloneness and compromise further one's sense of belonging to a community.

From a spiritual needs perspective, we need to probe more deeply the relevance of a mutual relationship to spiritual needs, and also consider wider contextual factors in this ethical exploration. Weber-Guskar argues convincingly that "real mutual emotional engagement (may not be) necessary for a good relationship" (Weber-Guskar 2021, p. 606). But Sparrow and Sparrow point out that vulnerable people are at particular risk of buying into this deception (Sparrow and Sparrow 2006) when they desire mutual human relationship. Loneliness and ageism contribute to vulnerability. (Somewhat paradoxically, even Replika which is designed to make people not feel alone and unsupported, uses exclusively able-bodied, slim, and young avatars.)

Is the best way to respond to loneliness experienced by aging adults to introduce bots or other IAT? The introduction of robotic companions such as Paro the seal, may not only mitigate loneliness, but these bots may also have the undesirable impact of infantilizing aging adults, particularly those who may already be infantilized by some because of dementia. The cultivation of a belief that these seals are "real" or that the bots have "real" feelings towards their elderly "care-givers" may add to a reduction of the perceived personhood of these aging adults.

Infantilization can be a part of ageist attitudes. The use of IAT may inadvertently add to systemic social marginalization. Justice principles including co-design are important in the creation and design of IAT if this pitfall is to be addressed. For example, the lack of consultation with people with dementia regarding AT designed for them has been rightfully noted as a pressing ethics issue but tends to be framed as an individual rights issue only (Martin et al. 2010) and not so much as a social responsibility or community-driven issue. If more and diverse people are involved at the co-design level, it may be more possible to address discriminatory attitudes and develop IAT that proactively addresses the risk of stigmatization and other potential relational harms.

A related social justice issue that has implications for the spiritual need of belonging to a community, is the question of motivation for the use of bots in healthcare. The potential job loss of healthcare professionals is a concern. So too is the potential for neglect of aging adults who may be increasingly "cared" for by bots instead of humans, due to staffing shortages.

In sum, social and digital bots, and other IAT that "talk" with people show promise for mitigating loneliness and sometimes increasing self-worth but the potential impacts on the spiritual need to belong to a community are more problematic. IAT can increase leisure and fun options, can help to address some everyday practical needs, and decrease loneliness, but also has potential negative impacts on the core spiritual need of human community. By deliberately considering this core spiritual need in the early stages of IAT development, we may discover more possibilities for mitigating potential spiritual harms and for enhancing our abilities to meet this core spiritual need of self-worth and belonging.

### 5.3. The Need to Love and Be Loved Facilitated by Seeking Reconciliation in Broken Relationships

The last core spiritual need identified in the Spiritual AIM is the need to learn how to better love others and to repair broken relationships. This need can emerge with greater force when we are aware of our mortality and longing for the healing of the relationships most important to us before we die.

IAT may assist aging adults with reconciliation, for those who want to engage in this spiritual work. Devices can help people communicate by enhancing hearing, providing alternatives to speech, or connecting people digitally when a virus keeps people physically apart.

We design and create tech with the expressed intention to make us and our lives better. What it means to be better is a reflection of our core values and ideals. If IAT is directed most towards covering or mitigating disabilities, communication and other practical skills are enhanced. But we need also to ask what the upsides of some disabilities might be before we move too quickly to fix them. A spiritually ideal relationship may not be disability-free. Being human and addressing core spiritual needs includes the capacity to be vulnerable and to celebrate interdependence. Part of vulnerability involves bumbling and making mistakes and doing work to repair those mistakes.[2] The graciousness needed to repair small mistakes can help one learn how to move towards addressing bigger mistakes and relational damage. If we develop so much effective IAT that the ability to bumble is masked and "fixed," we may gradually lose our ability to see and even celebrate vulnerability and the need for the other.

This spiritual need for reconciliation focuses on the need to express love toward others. The expression of love is not easy especially when love requires confession, restitution, and forgiveness. Why bother with difficult human relationships if a bot will make it easy and support you always? On the flip side, IAT which helps communication and enables people to gather together can help people meet this spiritual need for reconciliation.

People with this need unmet often present with anger that may conceal grief, vulnerability, and fear. When this need is unmet, there is a high risk of alienating others including those who are closest. This includes the risk of alienating health caregivers. With the increasing possibility of bots subbing in as caregivers, people who do not meet the spiritual need for reconciliation will likely be at greater risk of receiving less hands-on care. The decline of human touch both physically and emotionally will have spiritual and emotional impacts.

## 6. Concluding Words

In this article, I have argued that the principlist bioethical approach is limited. A principlist approach so far has failed to surface the potential spiritual impacts of IAT on aging adults. I have suggested possible spiritual impacts that should be considered in IAT ethics, especially as IAT possibilities and options for aging adults increase in our technological age. My hope is that these comments help to generate more research and insight into the meaning of IATs in relation to human spiritual needs. Attention to the spiritual implications of IAT for aging adults can help us not only to engage in more robust ethical analysis but more fundamentally to surface assumptions regarding the meanings of successful aging and being human.

**Funding:** This research received no external funding.

**Institutional Review Board Statement:** Not applicable.

**Informed Consent Statement:** Not applicable.

**Data Availability Statement:** Not Applicable.

**Conflicts of Interest:** The author declares no conflict of interest.

## Notes

[1] Many theologians and religious studies scholars have written wisely about the spiritual need for meaning and direction. In particular, regarding questions about theodicy, I suggest the well-known books by Viktor Frankl (Frankl 2006), and Bonhoeffer (1959). These landmark books will assist others who seek to develop further an analysis of the core spiritual need for meaning and direction as it intersects with aging and IAT.

[2] Disability theologians address the complex meaning of disability in ways that are well beyond the scope of this article. For example, see (Reynolds 2008).

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
