# Peer review of "Intelligent Assistive Technology Ethics for Aging Adults: Spiritual Impacts as a Necessary Consideration"

_religions, doi:10.3390/rel13050452_

Round 1

Reviewer 1 Report

I have no reservations about publishing this manuscript in Religions.

Further to this work, the author might consider the following comments:

  • Consider the issue of gerotranscendance, i.e. the fact that aging adults experience a different perception of the meaning of life. See works by Lars Tornstam and others. For instance, although this may seem paradoxical, the fear of loneliness appears to be higher in younger generations than among the ederly.
  • Thus, the understanding of spirituality might also change with time, with a shift from a materialistic and rational view of the world to a more cosmic and transcendent one. In this case, core spiritual and ethical values should be elaborated from surveys of aging adults rather than from younger people’s interpretation.
  • If perception of the meaning of life evolves in the course of life, one could wonder whether IAT might interfere with individual spiritual development, and not only with core spiritual needs that would be the same for everyone at any age. For instance, technological nudges might orientate people’s mindset so that they remain more materialistic than they would be otherwise.

Spelling corrections: see yellow highlighting in the appended file.

  • Line 260: specify the meaning of OT (Occupational Therapy)
  • Lines 418-419: check spacing
  • Lines 647-679: idem.

Author Response

Thank you very much for your careful reading of my article! I appreciated the observations you made throughout the PDF and have made all the changes you noted in the appended file.

I have added a mention of Tornstam’s theory under the section on meaning and direction. I have also made it clearer that many scholars have published very helpful work regarding theories on meaning, spirituality and aging. I agree that loneliness may be less of a core need for some aging adults. For others, as COVID has shown us, loneliness has exacerbated heath and well-being. Certainly, it would be worth investigating these questions in much more depth—perhaps in future articles. I see this article as merely introducing questions about spirituality and aging that are being neglected in IAT ethics discussions. I am starting to see that a book could (and should) be written in order to develop this intersection in much more depth. I think you’re right that the development of IAT is having impacts on the entire developmental span.

Thank you so much. Your observations have improved the article and have helped me to consider possible future topics for exploration.

Reviewer 2 Report

This is a well researched paper addressing a very important issue: the spiritual impact of IAT.  However, there are at least three areas that require further development. 

Section 5.1 on meaning never address the thought of Viktor Frankl, MD, PhD.  Frank has half a decade of scholarship on the role of meaning in psychiatry.  The paper should be revised to incorporate his thought.

Another lacuna is that there is no reference to cybernetics.  Many of the issues raised in the article have been examined by the second-order cybernetics community for years.  The authors should look into the literature of the cybernetics journals and books.  One place to start could be

Justin Anderson, Joseph R. Laracy, Thomas Marlowe, “A Multi-Disciplinary Analysis of Catholic Social Teaching with Implications for Engineering and Technology,” Journal of Systemics, Cybernetics, and Informatics 18, no. 6 (2020): 41-49.

and

Thomas Marlowe and Joseph R. Laracy, “Philosophy and Cybernetics: Questions and Issues,” Journal of Systemics, Cybernetics, and Informatics 19, no. 4 (2021): 1-23.

Finally, while there is frequent discussion of spirituality in the article there is no mention of the soul.  The authors should engage the question of what type of "relationship" can exist between an (ensouled) human person and an electronic assistive technology.  Answers depend on one's philosophical and theological anthropology.  It would be beneficial for the authors to engage ancient (Aristotle, Plato, Thomas Aquinas, etc.) and contemporary thinkers (Edward Feser, John Paul II, etc.).  For example, Aquinas argues that a human
being is a “hylomorphic” union of body and soul.

Author Response

Thank you for your thoughtful suggestions. I added mention of Frankl (and Bonhoeffer) in a footnote to help interested readers explore this core spiritual need further.

I agree that cybernetics intersect with IAT very much. While I do not explicitly use the term “cybernetics,” I have included mention of communication science in machines such as BCIs. Machine learning and algorithms have changed infomatics. I have made reference to these as important examples of IAT.

I also agree that a theological exploration of the soul could add even more to an exploration of the intersection of aging, spirituality and IAT. I think that this would make a great topic for a book. Unfortunately, I can only hope to stimulate other scholars, perhaps such as yourself, to consider such a book project. I agree that there are many topics that are hinted at in this article and beg book length explorations. I hope that the modest, limited scope of my article helps others to go forward into these topics. (As a side note, I have written elsewhere more about the integration of personhood including the soul, in relation to AI and human enhancement. I would love to add more to this article but it is already verging on 9,000 words.

Again, thank you for your thoughtful feedback. Your comments will help me in future projects and added to this article.

Reviewer 3 Report

This is a much-needed analysis of the potential of using AI and robotics to supplement the spiritual needs of patients. There are a few works that I would recommend to show fluency in the robotics and AI ethics community. 

  • Kate Darling, The New Breed: What Our Past With Animals Reveals About Our Future with Robots. 2021
  • Joshua K. Smith. Robot Theology: Old Questions through New Media. 2022. 
  • Eduard Fosch-Villaronga, Robots, Healthcare, and the Law. 2019

Author Response

Thank you so much for these sources! I have read Darling’s book but not the other two. I have ordered a copy of Smith’s book and I’m sure I will use it in future writing. I added mention of Smith’s book in a new sentence towards the end of the section on the spiritual need of meaning and direction. I am grateful that you suggested this.

I would very much like to add all these sources and more to this article but it is already over the recommended 8,000 words. I hope that my article can help stimulate the thinking of others and raise some awareness of the importance of spiritual impacts for IAT ethics.